# Enhanced Energy Density for P-Doped Hierarchically Porous Carbon-Based Symmetric Supercapacitor with High Operation Potential in Aqueous H_2_SO_4_ Electrolyte

**DOI:** 10.3390/nano11112838

**Published:** 2021-10-25

**Authors:** Xiaozhong Wu, Xinping Yang, Wei Feng, Xin Wang, Zhichao Miao, Pengfei Zhou, Jinping Zhao, Jin Zhou, Shuping Zhuo

**Affiliations:** 1School of Chemistry and Chemical Engineering, Shandong University of Technology, Zibo 255049, China; xiaozhongwu@yeah.net (X.W.); xinpingyang996@163.com (X.Y.); wang0126xin@163.com (X.W.); miaozhichao@sdut.edu.cn (Z.M.); pengfeizhou1231@126.com (P.Z.); jpzhao@sdut.edu.cn (J.Z.); zhoujin@sdut.edu.cn (J.Z.); 2Shandong Qilu Keli Chemical Institute Co., Ltd., Zibo 255086, China; Fengwei007@126.com

**Keywords:** hierarchically porous carbon, energy density, supercapacitor, current density, freeze-drying

## Abstract

Phosphorus-doped hierarchically porous carbon (HPC) is prepared with the assistance of freeze-drying using colloid silica and phytic acid dipotassium salt as a hard template and phosphorus source, respectively. Intensive material characterizations show that the freeze-drying process can effectively promote the porosity of HPC. The specific surface area and P content for HPC can reach up to 892 m^2^ g^−1^ and 2.78 at%, respectively. Electrochemical measurements in aqueous KOH and H_2_SO_4_ electrolytes reveal that K^+^ of a smaller size can more easily penetrate the inner pores compared with SO_4_^2^^−^, while the developed microporosity in HPC is conducive to the penetration of SO_4_^2−^. Moreover, P-doping leads to a high operation potential of 1.5 V for an HPC-based symmetric supercapacitor, resulting in an enhanced energy density of 16.4 Wh kg^−1^. Our work provides a feasible strategy to prepare P-doped HPC with a low dosage of phosphorus source and a guide to construct a pore structure suitable for aqueous H_2_SO_4_ electrolyte.

## 1. Introduction

As a new type of energy storage device, supercapacitors have attracted more and more attention in recent years. The durability of supercapacitors can be assessed by estimating their capacitance, equivalent series resistance, and impedance [1]. Among the components for supercapacitors, electrode materials and electrolytes play dominant roles in their capacitive performance [2,3,4,5]. Carbon materials with hierarchically porous structures are intensively studied due to their rational pore structures, good electric conductivity, and stability, which are beneficial when equipping carbon electrodes with different electrolytes. For instance, organic or ionic liquid electrolytes usually possess a high operation potential, which is decisive to the high energy density according to the following calculation formula for energy density: *E* = 0.5*CV*^2^, where *E*, *C* and *V* stands for energy density, specific capacitance, and operation potential for the assembled supercapacitor, respectively. However, the high viscosity and low ionic conductivity for organic or ionic liquid electrolytes often restrict the output power density of supercapacitors. By contrast, aqueous electrolytes with low viscosity and good ionic conductivity usually exhibit a high-power output [6]. As a typical aqueous electrolyte, neutral electrolytes with low concentration of H^+^ or OH^−^ ions can endure a wide operation potential over 1.5 V compared with acidic or alkaline electrolytes [7,8,9]. Recent studies on “water-in-salt” electrolytes, in which the salt concentration exceeds the water molecules both in weight and volume, have been explored to achieve an operation potential range of 2.2~3 V [10,11,12,13]. Such a high operation potential can be attributed to the generated electrode–electrolyte interphase between the high-concentration salt layer and carbon surface to lock down free water molecules with large anions, suppressing the chemical activity of water electrolysis [14,15].

Although acidic and alkaline electrolytes such as H_2_SO_4_ and KOH possess a higher conductivity than neutral electrolytes, the high concentration of H^+^ or OH^−^ ions is conducive to the hydrogen evolution or oxygen evolution reactions during electrochemical measurements, resulting in an operation potential within the theoretical decomposition potential of 1.23 V. Many strategies have been developed to expand the operation potential window. For instance, the conjunction of alkali and acid electrolytes via a K^+^-conducting Nafion membrane or Janus membrane can combine the low H_2_ evolution potential for the negative electrode and high O_2_ evolution potential for the positive electrode, realizing a high stable working potential of over 2 V [16,17]. Moreover, the operation potential window for aqueous electrolytes can be expanded by tuning the surface properties for carbon electrode materials. For instance, the introduction of oxygen or sodium functionalities on carbon cloth results in an expanded operation potential window for aqueous Na_2_SO_4_ electrolytes. The adsorbed Na^+^ on the carbon surface or Na-containing group can increase the onset overpotential for an oxygen evolution reaction or inhibit the adsorption of H^+^ in the electrolyte, thus widening the operation potential window [18,19]. When introducing N, O, P on porous carbon, the potential window of symmetric supercapacitor can be expanded to 1.5 and 1.9 V in aqueous KOH and Na_2_SO_4_ electrolyte, respectively [20]. It is widely accepted that N, O functionalities can bring about significant pseudocapacitance by reacting with acidic electrolytes. Therefore, developing heteroatom-doped porous carbon with a high operation potential in acidic electrolytes is of great importance. Ever since Hulicova-Jurcakova and coworkers found that the introduction of phosphorus in porous carbon can promote the operation potential for H_2_SO_4_ electrolyte due to the blockage of active oxidation sites on porous carbon by phosphate functionalities in 2009 [21], P-doped carbons have been prepared through a phosphoric acid-activation process and used as electrode materials to widen the operation potential for H_2_SO_4_ electrolyte [22,23,24]. Recently, theoretical modeling of P-enriched multi-heteroatom doped carbon electrodes shows that heteroatom doping can decrease the adsorption energy of carbon, thus storing extra cations per site and boosting the charge storage efficiently. Furthermore, the pre-adsorbed cations on heteroatom-doped carbon can significantly impede the hydrogen evolution reaction, giving rise to a wider operation potential window [25].

Even though the phosphoric-acid activation method can simultaneously introduce phosphorus functionalities during the activation process, the prepared porous carbons usually exhibit a microporous nature. Efforts on the preparation of P-doped carbon materials with hierarchically porous structures, which has been demonstrated to be effective for supercapacitors, should also be made. Herein, we prepare a phosphorus-doped hierarchically porous carbon (HPC) through a hard-templated method using colloid silica as the template with the assistance of the freeze-drying process. Material characterizations show that the prepared porous carbon exhibits a hierarchically porous structure with micro-, meso- and macro-pores and the Brunauer–Emmett–Teller (BET) specific surface area can reach up to 892 m^2^ g^−1^. CO_2_ sorption analysis shows that numerous ultramicopores (<0.6 nm) are present in HPC. Phosphorus prefers to be doped under a high temperature (800 °C). Electrochemical measurements in aqueous KOH and H_2_SO_4_ electrolytes reveal that HPCs with similar BET surface areas show quite different supercapacitive properties due to the different ionic sizes of K^+^ and SO_4_^2−^. Considering the detailed pore-structure parameters and ionic size in aqueous electrolyte, the developed microporosity in HPC8 is conducive to the penetration of SO_4_^2−^ compared with HPC6. Configurated with the P-doping and developed microporosity, the assembled symmetric supercapacitor can endure a high operation potential of 1.5 V and deliver a high energy density of 16.4 Wh kg^−1^ in aqueous H_2_SO_4_ electrolyte.

## 2. Experimental Section

### 2.1. Material Preparation

Dopamine (hydrochloride), colloid silica (30 wt% suspension in water), phytic acid dipotassium salt, and hydrofluoric acid (HF) were purchased from Sigma Aldrich and used as received without further purification. In a typical experiment, 0.36 g dopamine and 0.09 g phytic acid dipotassium salt were dissolved in 18 mL water by magnetic stirring to form a transparent solution, and then 2 mL colloid silica was added into the solution. The mixture was magnetically stirred for 10 min followed by being transferred into a plastic beaker and then immersed in liquid nitrogen and kept for another 10 min. The frozen mixture was then placed in a freeze-dryer and dried for 24 h. The dried product was further placed in a tube furnace and carbonized at a target temperature (600 and 800 °C) for 1 h with a heating rate of 2 °C min^−1^ under Ar flow rate of 50 cm^3^ min^−1^. The product was thoroughly washed with HF, deionized water, and ethanol to remove the silica and dried in a vacuum oven at 100 °C. The obtained porous carbons carbonized at 600 and 800 °C were labeled as HPC6 and HPC8, respectively. To investigate the effect of freeze-drying on the porous structure of carbon, the mentioned mixture was directly vacuum-dried at 50 °C instead of being frozen and freeze-dried, followed by carbonized at 800 °C. After the same post-treatment process, the obtained carbon sample was denoted as C8.

### 2.2. Material Characterization

The microstructure of the samples was characterized by X-ray diffraction (Rigaku SmartLab, Tokyo, Japan) using Cu Ka radiation and Raman spectroscopy (Wi-Tec micro Raman, Ulm, Germany) using a 532 nm wavelength laser. The morphologies of the samples were observed using a scanning electron microscope (SEM, FEI Verios 460, Hillsboro, OR, USA) equipped with an energy dispersive spectroscopy (EDS) detector and a transmission electron microscope (TEM, JEOL2100, Tokyo, Japan). The surface properties of the samples were characterized by X-ray photoelectron spectroscopy (XPS, K-Alpha, Thermo Fisher, Waltham, MA, USA). Physisorption analyses were performed with an ASAP 2020M analyzer (Micromeritics, Norcross, GA, USA) using nitrogen and carbon dioxide as adsorbates under 77 and 273 K, respectively. Pore size distribution was determined by QSDFT and NLDFT methods.

### 2.3. Electrochemical Measurements

The prepared HPCs were mixed with polytetrafluoroethylene in a mass ratio of 95:5 followed by pressed onto titanium mesh to obtain the working electrode. The active material in each electrode is about 2 mg. The employed electrolytes are 1 M H_2_SO_4_ and 6 M KOH aqueous solutions. All the electrochemical measurements such as cyclic voltammetry (CV), galvanostatic charge–discharge (GCD) and electrochemical impedance spectroscopy (EIS) were conducted on an IVIUM potentiostat (Ivium-n-Stat, Eindhoven, Netherlands). In a three-electrode system, Pt foil and saturated calomel electrode (SCE) were used as counter electrode and reference electrode, respectively. The potential window was set to be 0~1 and −1~0 V for acidic and alkaline electrolytes, respectively. The gravimetric specific capacitance (C_g_, F g^−1^) is calculated from the discharge curves according to the following equation: *C_g_* = *I*Δ*t*/*mV*, where *I*, Δ*t*, *m* and *V* are the discharge current, discharge time, mass of active material in each electrode and potential window, respectively. Symmetric supercapacitor cells with H_2_SO_4_ electrolyte were also assembled with two working electrodes. The operation potential was set to be 1 and 1.5 V for HPC8. Energy density (*E*, Wh kg^−1^) and power density (*P*, kW kg^−1^) of the cells could be calculated according to the equations: *C_cell_* = *I*Δ*t*/2*mV*, *E* = *C_cell_V*^2^/7.2, *P* = *E*/Δ*t*, where *C_cell_* is the specific capacitance for the supercapacitor.

## 3. Results and Discussion

Colloid silica was used as a hard template to prepare hierarchically porous carbon. As shown in Figure 1, the carbon precursor and phosphorus source were dissolved in colloid silica under magnetic stirring for 10 min. The obtained dispersion was frozen by liquid nitrogen followed by freeze-drying and further pyrolysis at target temperatures under Ar protection. The carbonized product was washed with HF acid to remove the silica template, obtaining the final HPC.

The prepared HPCs possess a sheet-like structure with a size of several tens of microns (Figure 1a and Appendix A). Uniform macropores with the size of several tens of nanometers to 300 nm can be observed on the surface of the sheet-like structure (Figure 1b,c)—compared with the SEM images for C8 (Appendix A) in which a flat surface with only a few macropores can be found. It can be inferred that the macropores in HPCs are generated due to the freeze-drying procedure. The water solvent was immediately frozen to solid ice during the liquid nitrogen freezing process. Furthermore, the freeze-drying process drives the sublimation of ice by leaving abundant macropores on the final carbon product. Apart from macropores, numerous mesopores with sizes of ~20 nm can be observed on TEM images (Figure 1d–g). These uniformly distributed mesopores are most probably derived from a silica template. Moreover, the relatively long-range parallel stacking of curved carbon layers could be observed as shown by arrows on the HR-TEM images (Figure 1e,g), indicating a relatively good crystallization for HPCs. Meanwhile, curved carbon layers (as marked in Figure 1e,g) give rise to nanoporosity, and serve as an active site for energy storage during electrochemical measurements. Elemental mappings for HPC8 show that apart from carbon elements, phosphorus, oxygen and nitrogen elements are uniformly distributed in this carbon sample (Figure 1h,k and Appendix A). The successive phosphorus doping was fulfilled, although very little phytic acid dipotassium was introduced as a phosphorus precursor. XPS analysis, to be discussed later, further confirms that there is a successive doping of phosphorus. Additionally, it shows an increase in P content as the carbonization temperature increases from 600 to 800 °C and the P content is detected to be 2.78 at% for HPC8 (Table 1).

Gas sorption analyses were employed to further detect the surface area and pore-structure parameters for HPCs. The adsorbed quantities (Figure 2a) of N_2_ for both samples can reach a plateau in the low relative pressure region (P/P_0_ < 0.01), which suggests the existence of micropores and small mesopores. Obvious hysteresis loops suggest the presence of mesopores in HPCs. The higher adsorbed quantity in the high relative pressure region (0.9 < P/P_0_ < 1.0) for HPC8 than that for HPC6 reveals a more developed porous structure for HPC8. In comparison, a relatively lower adsorbed quantity within the whole relative pressure region for C8 (Appendix A) suggests the lower content for micropores and mesopores. CO_2_ sorption analysis under a relatively high temperature (273 K) was used to probe the microporosity of HPCs. The adsorbed quantity (inset in Figure 2a) for HPC8 is also higher than that of HPC6 within the whole relative pressure region, indicating a more developed microporosity for HPC8.

The corresponding pore size distributions derived from N_2_ and CO_2_ sorption branches provide a better visualization of the porous structure (Figure 2b). The QSDFT pore size distribution shows typical micropore (~0.72 nm) and mesopore (15–30 nm) distribution and the latter one is inherited from silica template. The size of the mesopores coincides well with the TEM observations. Compared with HPC6, the increased peak intensities of the peaks centered at <1 nm and 20–30 nm for HPC8 reveal that high-temperature carbonization leads to the increase in these pores, contributing to the enlargement of pore volume. The specific surface areas and pore volumes for HPC8 and C8 (Table 2) confirm that the freeze-drying process can significantly promote the porosity of the carbon product. The NLDFT pore size distribution derived from CO_2_ sorption shows that numerous ultramicropores smaller than 0.6 nm can be detected by the CO_2_ probe. Moreover, it is obvious that HPC8 shows a higher content of ultramicropores than that of HPC6. The tabulated surface area and pore-structure parameters for HPCs (Table 2) suggest that even though HPCs have close BET specific surface areas, the micropore surface area determined by the CO_2_ probe for HPC8 is much larger than that of HPC6.

The XRD patterns of HPCs (Figure 3a) exhibit a distinct peak centered at 2θ = 21.2°, suggesting a relatively good graphitization degree for HPCs. Such a result can coincide well with the HR-TEM observation (Figure 1e,g). The calculated d_002_ for HPCs is about 0.418 nm. The higher d002 value compared with that of graphite is probably due to the heteroatom doping, which expands the interlayer distance between adjacent carbon layers. Two distinct peaks in the Raman spectra for HPCs (Figure 3b) at 1351 and 1589 cm^−1^ are assigned to the D and G bands for carbon materials, respectively. The D band is assigned to the disorder-induced mode associated with structural defects and imperfections while the G band is assigned to the first-order scattering of the E2g mode from the sp^2^ carbon domains [26]. The intensity ratio I_G_/I_D_ is used as a measure of the graphitization degree for carbon samples. Both HPCs possess an I_G_/I_D_ value higher than 1, 1.11 for HPC6 and 1.09 for HPC8, indicating a good graphitization degree for HPCs. The XPS analyses show that phosphorus and nitrogen are doped in HPCs (Figure 3c). The N 1s spectrum could be deconvolved into three peaks at 398.1, 400.5 and 403.3 eV, which could be assigned to pyridinic (N1), pyrrolic (N2) and pyridine-N-oxide (N3) nitrogen species, respectively [27]. The P 2p spectrum could be deconvolved into three peaks at 131.2, 132.8 and 134.3 eV, corresponding to P-C bonding (P1), pyrophosphate ([P_2_O_7_]^4−^, P2) and metaphosphate ([PO_3_]^−^, P3) species, respectively [22]. The contents of N and P (Table 1) in HPCs suggests that nitrogen would escape under a high-temperature treatment while P prefers to stay under a high temperature.

Electrochemical measurements were systematically employed in 1 M H_2_SO_4_ and 6 M KOH electrolytes in a three-electrode system. It is shown that HPC8 exhibits a much higher response current density than that of HPC6 during CV measurements in both electrolytes (Figure 4a, Appendix A and Appendix A). Additionally, obvious redox peaks could be found in the CV curves for HPC8, suggesting the presence of pseudocapacitance, which is attributed to the faradic reactions between surface functionalities and electrolytes. The deviation from a linear shape for the GCD curves (Figure 4b, Appendix A and Appendix A) further confirms the presence of pseudocapacitance. The current density dependence of specific capacitance (Figure 4c and Appendix A) shows that HPC8 possesses a much higher specific capacitance than that of HPC6 regardless of whether in a KOH or H_2_SO_4_ electrolyte. The corresponding areal capacitance for HPC8 is much larger than that of HPC6 in both electrolytes (Table 3), suggesting an improved usage efficiency for the available surface area to electrolyte. Additionally, HPCs show higher areal capacitance in H_2_SO_4_ electrolyte than that in KOH electrolyte, which is because the surface oxygen/nitrogen functionalities prefer to introduce pseudocapacitance in acidic electrolyte. However, each HPC sample exhibits a higher capacitance retention ratio (Appendix A) in KOH electrolyte than that in H_2_SO_4_ electrolyte. Such a trend can be attributed to the difference in ionic size and porous structure for HPCs. K^+^ and SO_4_^2−^ are preferentially adsorbed on the electrode/electrolyte interface in the negative and positive potential ranges in KOH and H_2_SO_4_ electrolyte, respectively. Additionally, K^+^ possesses a smaller ionic size (0.36–0.42 nm) than that of SO_4_^2−^ (>0.58 nm) [28]. Our previous study reveals that ultramicropores are able to be penetrated by K^+^ [29,30]. Therefore, the high capacitance retention ratio in KOH electrolyte for each HPC is because it is easier for K^+^ to penetrate the inner pores. As discussed above, HPC8 shows an obviously higher micropore volume with the pore size <1 nm. In this case, an enhanced supercapacitive performance in H_2_SO_4_ electrolyte for HPC8 is achieved in comparison with HPC6.

To investigate the chemical and physical processes on the interface of electrode and electrolyte, electrochemical impedance spectroscopy was carried out and the corresponding Nyquist plot is shown in Figure 5a. The intercept with the real axis of the plot indicates the equivalent series resistance (ESR) value, which is known as the internal resistance, defined as the sum of the resistances of the electrode, electrolyte and the contact resistance between the electrode and current collector. Considering that the same current collector and electrolyte are used in this work, the ESRs could be a reference for the intrinsic resistance of carbon materials. The ESRs for HPCs were found to be 0.51 (HPC8) and 0.60 Ω (HPC6), revealing a good electrical conductivity for both samples. At the medium-high frequency region, the diameter of the semicircle represents the charge transfer resistance (R_ct_), which is correlated with the porous nature of carbon in accordance with electrolyte-accessible area and electrical conductivity. The absence of a semicircle for HPC6 reveals that electron transfer probably occurs within the large fraction of wide pores (mesopores) that have easy accessibility and minimal charge [31]. Such a result further confirms that electrolytes cannot penetrate the inner pores of HPC6, resulting in a low surface usage efficiency for HPC6. The equivalent circuit and fitted resistances for HPC8 can be found in Appendix A. At the low frequency region, the slope for HPC8 is close to 90° along the -Z”, suggesting an EDL-dominated capacitive nature. The deviation from the straight line along the -Z” for HPC6 indicates a nonideally polarizable electrode. Figure 5b shows the dependence of capacitance on the frequency for HPCs. The capacitance gradually decreases with the increase in frequency, suggesting that the electrolyte ions are unable to penetrate the surface of electrode at high frequency. The capacitance values for HPC8 at low frequency (0.01 Hz) are much higher than that of HPC6, which coincides well with the result of GCD at the current density of 1 A g^−1^. In comparison with the capacitance variation trend of HPC8, the capacitances for HPC6 do not show signs of saturation at the low frequency of 0.01 Hz, suggesting that equilibrium ion adsorption is not achieved within 100 s. The relationship between phase angle and frequency (Figure 5c) shows that the phase angle for HPC8 is −82°, while the value for HPC6 is −50°, suggesting a better EDL-dominated capacitive nature for HPC8. It is shown that the frequency-dependent impedance decreases with the increase in frequency for all samples (Figure 5d). Typically, HPC8 possesses a lower impedance compared with HPC6, which is beneficial for ionic diffusion from the electrolyte to porous electrode, resulting in a good supercapacitive performance for HPC8.

As for P-doped carbons, it was reported that the active oxidation sites on the surface of porous carbon can be blocked by phosphate functionalities, resulting in an operation potential higher than 1.5 V. Considering that HPC8 with a relatively high P content shows a perfect supercapacitive performance in H_2_SO_4_ electrolyte, an HPC8-based symmetric supercapacitor was assembled and tested under the operation potentials of 1 and 1.5 V in 1 M aqueous H_2_SO_4_ electrolyte. The Nyquist plot for the HPC8-based symmetric supercapacitor (Appendix A) shows a low internal resistance of 1 Ω. CV curves (Figure 6a) at a low operation potential show a pair of redox peaks, suggesting the energy storage mechanism of EDL capacitance and pseudocapacitance, which is generated by the redox reaction between H^+^ and surface functionalities [32]. However, the absence of redox peaks for CV curves (Figure 6a and Appendix A) measured under a 1.5 V operation potential window suggests that the EDL capacitance dominated the energy storage mechanism under such a high operation potential window. The overlapped GCD curves (Figure 6b and Appendix A) measured at different operation potentials reveal a good reversibility for HPC8. The energy density for the 1.5 V supercapacitor can reach up to 16.4 Wh kg^−1^, which is much larger than that (6 Wh kg^−1^) of the 1 V supercapacitor. Moreover, with the hierarchically porous structure, the energy density can be retained at 9 Wh kg^−1^, even at a high power output of 7500 W kg^−1^. The cyclic stability and CV curves at different cycles for the HPC8-based symmetric supercapacitor (Appendix A) suggest the EDL capacitance-dominated energy storage mechanism of HPC8. The inadequate durability for such carbon is probably due to its developed porous structure.

## 4. Conclusions

Hierarchically porous carbons with P-doping were prepared through a hard-templating method with the assistance of the freeze-drying process, which leads to a developed porosity for HPC. A low-dose phosphorus source (phytic acid dipotassium salt) can ensure a successive P-doping, and P atoms prefer to be stably doped at a high carbonization temperature of 800 °C. Electrochemical measurements in KOH and H_2_SO_4_ electrolytes reveal that K^+^ more easily penetrates the inner pores compared with SO_4_^2−^ due to their ionic size difference, whereas a more developed microporosity in HPC is conducive to the penetration of SO_4_^2−^. Moreover, P-doping in HPC leads to a stable operation potential up to 1.5 V in H_2_SO_4_ electrolyte for the assembled symmetric supercapacitor. Our work provides a feasible strategy to prepare P-doped HPC for a supercapacitor with a high operation potential and a guide to construct a pore structure suitable for aqueous H_2_SO_4_ electrolyte.

## Data Availability

Data are contained within the article.

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
