# Peer review of "Enhanced Energy Density for P-Doped Hierarchically Porous Carbon-Based Symmetric Supercapacitor with High Operation Potential in Aqueous H2SO4 Electrolyte"

_nanomaterials, 2021, doi:10.3390/nano11112838_

Round 1
Reviewer 1 Report
The Manuscript ID “nanomaterials-1388851” titled “Enhanced Energy Density For P-doped Hierarchically Porous Carbon-Based Symmetric Supercapacitor with High Operation Potential in Aqueous H2SO4 Electrolyte” describes the preparation of Phosphorus-doped hierarchically porous carbon (HPC) with the assistance of freeze-drying using colloid silica and phytic acid dipotassium salt as hard template and phosphorus source respectively and employed as supercapacitor electrode using KOH and H2SO4 electrolyte. After careful evaluation, this reviewer found major issues in this manuscript. Therefore, I recommend the Editor to accept this paper after addressing the following comments.
.
The issues found in the manuscript are:
- Cyclic voltammetry graphs of HPC8 at different scan rates should be attached.
- Charge discharge graphs of HPC8 at different current density should be attached.
- Why did they use different molarity of acidic and alkaline electrolyte?
- Circuit fitting data of Nyquist plot should be attached.
- Cyclic voltammetry graphs of the symmetric device at different scan rates should be attached.
- Charge discahrge graphs of the symmetric device at different current density should be attached.
- Cyclic stability of the symmetric device should be attached.
- Electrochemical impedance spectroscopy analysis of the symmetric device should be plotted.

Author Response
We thank the reviewer for his/her comments to improve our manuscript. A point-by-point response is attached. The manuscript and supporting information have also been revised according to the comments.

Reviewer 2 Report
This is interesting manuscript presenting a way of increasing operating voltages in the proposed super capacitors. I have found only two remarks that could improve the manuscript:
- Please, give some comments about pore size distribution (Fig. 2) and its impact on durability and capacitance.
- The presented references and comments don't mention anything about durability of such construction and possibility of further and necessary investigations in this area. Please add some references about possible ways of checking their quality, e.g., Szewczyk, A. (2017). Measurement of noise in supercapacitors. Metrology and Measurement Systems, 24(4).
Author Response

(The authors gave the same response as above.)

Round 2
Reviewer 1 Report
The manuscript has been revised accordingly.